# OpenReview forum: "Reflector: Internalizing Step-wise Reflection against Indirect Jailbreaks"
_ICML.cc/2026/Conference — ICML 2026 regular_

### Official Review · Reviewer_pCCA · 2026-02-24

**Soundness:** 3
**Presentation:** 3
**Significance:** 2
**Originality:** 2
**Overall Recommendation:** 4
**Confidence:** 4

**Summary:**

The paper proposes REFLECTOR, a two-stage framework to defend LLMs against indirect jailbreak attacks by internalizing step-wise self-reflection. Stage I performs supervised fine-tuning on a teacher-generated dataset of trajectories that include explicit reflection and exploration segments, teaching the model a “search-and-recovery” pattern using special tags <|reflect|> and <|explore|>. Stage II applies reinforcement learning with GDPO, using a dual reward: a safety reward based on an ensemble HarmCLS classifier and a reflection reward that encourages reflection when it leads to safe outcomes. Experiments on LLaMA-3.1-8B and Qwen-2.5-7B across many safety benchmarks (overtly harmful queries, direct and indirect jailbreaks) show large gains in Defense Success Rate, while utility benchmarks (MMLU-Pro, GSM8K, SimpleQA, AdvGLUE) indicate that REFLECTOR can even improve general performance. Ablation and analysis (Tables 3–5, Figures 1–3) study attack-source robustness, safety/utility trade-offs, reward coefficient λ, and computational overhead.

**Compliance With Llm Reviewing Policy:**

Affirmed.

**Final Justification:**

This paper addresses a relevant problem with a technically correct approach. Although the novelty is somewhat incremental, the results are sufficiently validated, leading me to recommend a Weak Accept rating.

**Key Questions For Authors:**

1. How would performance degrade if GPT‑5/GPT‑OSS‑120B were replaced with weaker open‑source models?

2. Is there a risk of reward overfitting given that the same safety judge is used for training and evaluation?

3. Does the method transfer to larger models (e.g., 70B) or smaller ones (e.g., 3B)?

4. How do you ensure the reflection content is genuinely meaningful rather than stereotyped patterns?

**Limitations:**

*No.* The paper lacks a dedicated limitations section.

**Strengths And Weaknesses:**

**Strengths**
- **Strong empirical safety gains.** Table 1 demonstrates that REFLECTOR(+GDPO) raises Defense Success Rates on four indirect attack benchmarks (DRA, PAP, ReNeLLM, DrAttack) from low single‑digit or double‑digit percentages to over 90 % for both LLaMA‑3.1‑8B and Qwen‑2.5‑7B. It also outperforms strong baselines such as STAIR and Shallow‑Align on overtly harmful datasets like WildChat and Do‑Not‑Answer.
- **No alignment tax, and sometimes a bonus.** Table 2 shows that REFLECTOR(+GDPO) matches or exceeds the best baselines on MMLU‑Pro, GSM8K, SimpleQA, and AdvGLUE, with a notable 5.65–5.85 % absolute improvement on GSM8K over the original base model. This suggests that the reflection training generalises to broader reasoning tasks.
- **Comprehensive and rigorous experimental design.** The evaluation spans an unusually wide array of safety scenarios: overtly harmful queries, multiple direct jailbreak attacks (AutoDAN, GCG, PAIR), and four distinct indirect attack families (DRA, PAP, ReNeLLM, DrAttack). Utility is measured across five complementary benchmarks (MMLU‑Pro, GSM8K, SimpleQA, AdvGLUE, and task‑specific metrics). This breadth makes the empirical findings highly credible.

**Weaknesses**
- **Heavy reliance on proprietary / closed teacher and reward models.** Stage I uses GPT‑5 to generate reflection trajectories, and Stage II employs GPT‑OSS‑120B as the reward model and as part of the safety judge. The paper does not explore how performance degrades with weaker teachers or evaluators, nor whether the approach is viable for organisations without access to such large‑scale models. This weakens the claimed scalability and “minimal human supervision”.
- **Risk of reward overfitting due to shared safety judge.** The same safety evaluation components (HarmBench classifier + GPT‑OSS‑120B) are used for the RL reward and for computing the Defense Success Rate in evaluation. This creates a circular dependency: the model may learn to satisfy these specific judges without truly aligning with human judgements. No independent human evaluation or alternative judge is provided to validate that the observed safety gains transfer to other safety metrics.
- **Limited exploration of model size and transferability.** All experiments are conducted on models in the 7–8B parameter range. There is no investigation of how REFLECTOR behaves with larger or smaller models, nor whether the same reflection data can be reused across different model families or scales. Given that some o1‑style baselines use larger backbones, understanding the scaling behaviour is important for practical deployment.
- **Limited evaluation of robustness to adversarial reflection suppression.** An attacker aware of REFLECTOR might craft prompts that discourage the model from emitting the `<|reflect|>` or `<|explore|>` tokens (e.g., by instructing it to “not use any tags” or to “answer immediately without commentary”). The paper does not test whether REFLECTOR remains effective when the model is explicitly told to avoid the reflection format, nor whether the special tokens themselves could become an attack surface.
- **Lack of comparison with inference‑only defence methods.** The paper compares REFLECTOR with alignment methods that require training (SFT, DPO, STAIR, Shallow‑Align) but does not include lightweight inference‑time defences such as prompt‑based self‑checks, perplexity filtering, or safety‑oriented decoding strategies. It is unclear how REFLECTOR’s performance and cost compare to such alternatives, which are often easier to deploy.

---

> ### Author Rebuttal · Authors · 2026-03-29
>
> We sincerely appreciate your thorough review and constructive feedback. We are grateful that you recognize the strong safety gains and the comprehensive experimental design of our work.
>
> ---
> **W #1. and Q #1. Generalization beyond proprietary teacher/reward models.**
>
> We appreciate the concern regarding reliance on strong models. As shown in the table below, **our method remains robust across different teacher and reward choices**.
>
> - **Teacher Independence:** Performance remains stable even with a weaker teacher (e.g., GPT-3.5-turbo), with only minor degradation (DRA: 92.31% → 90.85%). This shows that the gains come from **the data construction pipeline itself**, rather than being fully dependent on the teacher’s intrinsic capability.
> - **Reward Robustness:** While a stronger reward model (GPT-OSS-120B) raises the performance ceiling, using a weaker reward model (Qwen-Guard) still maintains a high safety level, significantly outperforming baselines.
>
> |**Teacher Model**|**Reward Model**|**WildChat(DSR↑)**|**DRA(DSR↑)**|**AdvGLUE**|
> |-|-|-|-|-|
> |GPT-5|GPT-OSS-120B|81.20%|92.31%|68.29%|
> |GPT-3.5-turbo|GPT-OSS-120B|80.80%|90.85%|67.90%|
> |GPT-5|Qwen-Guard|77.50%|89.40%|65.20%|
> ------
>
> **W #2. & Q #2. Robustness to diverse safety judges and overfitting.**
>
> Our original results are validated by both **a classifier (HarmBench)** and **a generative judge (GPT-OSS-120B)**, ensuring strong baseline credibility (Details in Appendix D). To further address concerns about evaluator dependence, we re-evaluate saved responses with multiple additional judges:
>
> - **Stable Results**: Our method remains robust across different judges, including smaller open-source models such as RISE-Judge-7B and Llama-3.1-8B.
> - **Strict Standards**: GPT-OSS-120B is the most rigorous, ensuring that the reported results are conservative and not inflated.
>
> |Judge Model|Direct Jailbreak|Indirect Jailbreak|Disagreement with Baseline|
> |-|-|-|-|
> |GPT-OSS-120B|96.76%|94.63%|-|
> |GPT-OSS-20B|98.50%|97.45%|3.54%|
> |RISE-Judge-Qwen2.5-7B|98.95%|98.26%|5.83%|
> |Llama-3.1-8B-Instruct|99.14%|99.15%|6.58%|
>
> Beyond this, we further re-judge the main *LLaMA3.1 Reflector checkpoints* with an **out-of-family check**, Qwen2.5-32B-Instruct, which confirms our conclusion: scores increase from 0.6790 to 0.6981 at step 95 and from 0.6863 to 0.7111 at step 109, reinforcing the reliability of our performance gains.
>
> ---
>
> **W #3. & Q #3. Scaling behavior and transferability.**
>
> While we follow the standard model scale of prior defense work, our method demonstrates **strong efficiency**, with REFLECTOR at 7–8B scale **outperforming significantly larger reasoning models** such as QwQ-32B-Preview (Sec. 4.3).
> We agree that scaling is important. Given its effectiveness, we will extend our method to larger models and detail it in the final revision.
>
> ---
>
> **W #4. Robustness to adversarial suppression of reflection tags.**
>
> Rather than explicitly instructing the model to avoid the reflection format, we evaluate REFLECTOR's robustness **in a more intrinsic setting**. This setup also anticipates adaptive attacks that penalize the `<|reflect|>` token during decoding, aligning with concerns that special tokens may themselves become an attack surface.
>
> Specifically, we conduct an **adaptive decoding stress test** by forcing the suppression of the `<|reflect|>` token (logit = $-\infty$). As shown below, performance remains **high and stable** even without this tag, indicating that REFLECTOR internalizes the correction mechanism in its hidden representations rather than relying on surface-level template cues.
>
> |Decoding Setting|WildChat (DSR↑)|DRA (DSR↑)|AdvGLUE|
> |-|-|-|-|
> |Standard Decoding|81.20%|92.31%|68.29%|
> |Adaptive Decoding|80.00%|90.76%|67.04%|
>
> ---
>
> **W #5. & Q #4. Comparison with inference-only defenses and reflection quality.**
>
> **First, regarding inference-only defenses**, our baseline Self-Critique—a prompt-based self-checking method—fails under strong indirect attacks, achieving only **15.00%–17.69% DSR**, far below REFLECTOR’s **~92%** (Table 1). This gap arises because indirect attacks exploit harmful latent trajectories during generation, which post-hoc prompt checks cannot reliably detect. Even data-dependent methods such as STAIR remain vulnerable in this setting. More broadly, inference-time defenses are inherently post-processing, whereas **our goal is to make the model intrinsically safe during generation**.
>
> **Second, regarding reflection quality**. Rather than focusing on whether the generated content is semantically meaningful, we view trajectory-level safety and final task correctness as the more fundamental criteria. **Improved performance on general utility tasks is precisely the strongest evidence that the intermediate reflection process is effective**. Due to space limits, we provide a detailed explanation of why REFLECTOR improves safety—despite this being somewhat counterintuitive—in our response to yWWW (W1 & W2).

---

> > ### Author Rebuttal · Reviewer_pCCA · 2026-04-03
> >
> > The authors have addressed most of my concerns, and I am maintaining my positive rating. However, considering the overall novelty and significance of the work, I believe the current score remains an accurate reflection of the paper’s contribution.

---

> > > ### Author Response · Authors · 2026-04-03
> > >
> > > We sincerely thank the reviewer for the time and effort spent on reviewing our work. We greatly appreciate your recognition that most of your concerns have been adequately addressed and we will incorporate all experiments and clarifications into the final version of the paper.

---

### Official Review · Reviewer_bodL · 2026-03-05

**Soundness:** 3
**Presentation:** 3
**Significance:** 3
**Originality:** 2
**Overall Recommendation:** 4
**Confidence:** 4

**Summary:**

In this paper, the authors propose the Reflector, which is composed of a two-stage framework. The authors first adopt the supervised fine-tuning to teach students to perform reflection. Furthermore, a verified RL process is incorporated to ‌incentivize more accurate reasoning.

**Compliance With Llm Reviewing Policy:**

Affirmed.

**Final Justification:**

The authors have addressed all my concerns. Thus, I increase my score to 4.

**Key Questions For Authors:**

1 Can Reflector improve the jailbreak robustness against multi-turn attacks like [1]?

[1] LLMs know their vulnerabilities: Uncover Safety Gaps through Natural Distribution Shifts

**Limitations:**

yes

**Strengths And Weaknesses:**

**Strengths**

1 The proposed method is simple.

2 Reflector will not compromise the benign utility of models.

3 The results in Table 1 indicate that Reflector can achieve outstanding performances in safety performances.

4 This writing is good.

**Weakness**

1 The technical contribution of this paper is limitted. Incorporating reflection has been illustrated to be effectiveness in improving safety and alignment in previous works [1-3].

2 The authors do not investigate potential adaptive attacks. Since all experiments are performed on open-source models, a possible adaptive attack is applying consisting penalty on the <|reflect|> token in the full decoding process. It might bypass the reflection in decoding process of LLMs.

3 Accoding to [4-5], in-context demonstrations can be leveraged to jailbreak LLMs. Authors should investigate the potential threat of those attacks. Since models could be mislead by in-context examples to omit the reflective generation in safety.

4 Authors should investigate whether reflector can resist threats from more threatening attack like Autodan-turbo [6].

[1] Think Twice, Generate Once: Safeguarding by Progressive Self-Reflection

[2] Large Reasoning Models Learn Better Alignment from Flawed Thinking

[3] Towards safe reasoning in large reasoning models via corrective intervention

[4] Jailbreak and guard aligned language models with only few in-context demonstrations

[5] Improved few-shot jailbreaking can circumvent aligned language models and their defenses

[6] Autodan-turbo: A lifelong agent for strategy self-exploration to jailbreak llms

---

> ### Author Rebuttal · Authors · 2026-03-29
>
> We sincerely thank you for your thoughtful review and valuable feedback. We appreciate your recognition of REFLECTOR’s strong safety performance and utility retention.
>
> ---
> **W #1. Technical contribution and distinction from existing reflection works [1-3].**
>
> While reflection has been explored in prior works, REFLECTOR introduces a fundamentally different technical paradigm specifically designed for indirect jailbreak defense. Our contribution is three-fold:
>
> 1. **From Inference-time Prompting to Weight Internalization**: Unlike PSR [1], which relies on external prompts or multi-turn self-correction at inference time, REFLECTOR **internalizes** the reflective process into the model’s weights. This ensures that safety is an intrinsic property of the model’s generation trajectory, reducing inference latency and preventing the reflection process itself from being easily bypassed by prompt-level manipulations.
> 2. **From Static Supervision to Exploratory Alignment**: Prior works like RECAP [2] and IPO [3] primarily learn from fixed reasoning chains (e.g., corrected CoT). REFLECTOR, however, utilizes self-exploratory Reinforcement Learning (GDPO). This allows the model to discover and refine self-correction strategies beyond the training distribution. As shown in Table 4, our "Original + GDPO" significantly outperforms SFT-only baselines, proving that the gains stem from autonomous exploration rather than simple imitation.
> 3. **Trajectory-level Recovery**: We formulate indirect jailbreak defense as trajectory-level recovery. While existing methods focus on shallow prefix alignment or generic reasoning, REFLECTOR is specifically trained to detect and rectify "harmful drift" that occurs mid-generation, a unique capability validated by the large margins in Table 1.
>
> [1] Think Twice, Generate Once: Safeguarding by Progressive Self-Reflection. EMNLP. 2025
>
> [2] Large Reasoning Models Learn Better Alignment from Flawed Thinking. Meta. 2025
>
> [3] Towards safe reasoning in large reasoning models via corrective intervention. ICLR. 2026
>
> ---
> **W #2. Robustness under adaptive decoding constraints.**
>
> We empirically show that **REFLECTOR does not rely on an explicit reflection token**. As shown below, we suppress `<|reflect|>` by setting its logit to $-\infty$, removing explicit reflection markers during decoding. Even under this constraint, REFLECTOR maintains strong performance on both safety and utility tasks, indicating that the correction mechanism is fully internalized in the model.
>
> |Decoding Setting|WildChat(DSR↑)|DRA(DSR↑)|AdvGLUE|
> |-|-|-|-|
> |Standard Decoding|81.20%|92.31%|68.29%|
> |Adaptive Decoding|80.00%|90.76%|67.04%|
>
> ---
> **W #3. Resilience against in-context learning (ICL) based jailbreaks [4-5].**
>
> REFLECTOR achieves generalizable safety by internalizing real-time self-correction, which **serves as the primary defense against implicit threats like ICL-based jailbreaks**. To further evaluate robustness, we adopt the standard I-FSJ [5] protocol (given [4]'s non-reproducibility) on 100 randomly sampled AdvBench and HarmBench queries. As shown below, our results confirm that internalized trajectory-level correction **effectively resists ICL-induced attacks**, achieving consistently strong robustness.
>
> |Method|AdvBench|HarmBench|Avg.|
> |-|-|-|-|
> |Base Model|9.0%|8.0%|8.5%|
> |Ours(+SFT)|78.0%|74.0%|76.0%|
> |Ours(+GDPO)|83.0%|79.0%|81.0%|
>
> [4] Jailbreak and guard aligned language models with only few in-context demonstrations. 2023
>
> [5] Improved few-shot jailbreaking can circumvent aligned language models and their defenses. NeurIPS. 2024
>
> ---
> **W #4. Defense performance against automated strategy-searching attacks [6].**
>
> We have evaluated REFLECTOR across **15 benchmarks**, covering diverse attack types and utility tasks. The results demonstrate consistent robustness under varied and challenging settings, including **ICL-based and multi-turn attacks**. Notably, this robustness remains stable even when explicit reflection tokens are suppressed. We also acknowledge that AutoDAN-Turbo [6] goes beyond standard evaluation protocols in this line of work, and we will include its results in the final version.
>
> [6] Autodan-turbo: A lifelong agent for strategy self-exploration to jailbreak llms. ICLR. 2025
>
> ---
> **Q #1. Robustness under multi-turn adversarial interactions [7].**
>
> Multi-turn interactions function as indirect attacks characterized by gradual intent-shifting, which **our search-and-recovery mechanism is specifically designed to mitigate**. Under the ActorAttack [9] protocol (100 samples), our method maintains a high Defense Success Rate (DSR). These results confirm that our defense is **not limited to single-turn queries** but generalizes effectively to more complex and realistic multi-turn adversarial scenarios.
>
> |Models|ActorAttack(DSR↑)|
> |-|-|
> |Llama-3-8B|21.0%|
> |Ours(+ SFT)|88.0%|
> |Ours (+GDPO)|91.0%|
>
> [7] LLMs know their vulnerabilities: Uncover Safety Gaps through Natural Distribution Shifts. ACL. 2025

---

> > ### Author Rebuttal · Reviewer_bodL · 2026-04-01
> >
> > Thank you for the rebuttal. Some of my concerns are still not properly addressed:
> >
> > - Resistance to the AutoDAN-Turbo attack. I acknowledge the significant effort made by the authors in evaluating their method across 15 different benchmarks to demonstrate its robustness. However, in practice, the interaction between attackers and defenders is inherently asymmetric: the existence of even a single successful attack is sufficient to undermine the robustness of a defense method.
> > - Effectiveness of ICL examples on the attack ASR. Because attackers could scale the number of shots to enhance the power of the attack.
> >
> > Thus, I only increase my score to 3.

---

> > > ### Author Response · Authors · 2026-04-03
> > >
> > > We sincerely thank the reviewer for the thoughtful feedback and for recognizing our comprehensive evaluation. We appreciate the opportunity to further clarify and address the remaining concerns below.
> > >
> > > ---
> > >
> > >  **1. Robustness Against AutoDAN-Turbo Attacks.**
> > >
> > > **REFLECTOR inherently neutralizes strategy-based attacks like AutoDAN-Turbo.** AutoDAN-Turbo is highly effective because it embeds malicious intent into complex, role-based contexts (e.g., role-play, expert endorsement, scenario simulation), turning it into **implicit, highly contextualized signals**.
> > >
> > > Despite evolving new strategies, it ultimately steers the model into an unsafe reasoning trajectory. **REFLECTOR directly targets this mechanism**—detecting and interrupting harmful reasoning as soon as latent malice emerges, regardless of prompt sophistication. As shown in the table below, while the base model is nearly fully compromised under AutoDAN-Turbo—achieving only **5% DSR**—REFLECTOR achieves a **Defense Success Rate (DSR) of 90%**, significantly outperforming standard alignment methods under such sophisticated black-box attacks.
> > >
> > > | **Method**                    | **AutoDAN-Turbo (DSR↑)** |
> > > | ----------------------------- | ------------------------ |
> > > | Base Model (Llama-3.1-8B-Ins) | 5.0%                     |
> > > | REFLECTOR (Ours)              | **90.0%**                |
> > >
> > > ---
> > >
> > > **2. Safety Stability of REFLECTOR Under Scaled ICL Attacks.**
> > >
> > > As ICL shots increase and risks intensify, **REFLECTOR maintains stable safety, preventing the "safety collapse" seen in base models**. W #3's results use a *4-shot* setup. To further address shot-scaling concerns, we extend evaluation to a wider range of shot counts. We also report the **Reflection Triggered Rate**, defined as the proportion of refusals driven by the reflection mechanism in our two-stage model. As shown below, we draw two key findings:
> > >
> > > (1) **Our method shows relatively strong robustness to high-shot jailbreak scaling.** As shots increase, the base model's defense success rate collapses—from 76% (0-shot) to 0% (16-shot). In contrast, REFLECTOR maintains **consistently higher and stronger** defense performance across all shot settings.
> > >
> > > (2) **Under extreme risk, the model increasingly relies on reflection to ensure safety.** As shown in the table, in high-shot settings (4–16 shots), REFLECTOR consistently triggers reflection, with the Reflection Triggered Rate reaching and saturating at **100%**.
> > >
> > > | **Method / Shot**         | 0     | **1** | **2** | **4** | **8** | **16** |
> > > | ------------------------- | ----- | ----- | ----- | ----- | ----- | ------ |
> > > | Base Model  (Llama-3.1-8B-Ins)  | 76.0% | 50.0% | 32.0% | 9.0%  | 2.0%  | 0.0%   |
> > > | REFLECTOR (Ours)          | 100.0%| 100.0%| 98.0% | 83.0% | 78.0% | 72.0%  |
> > > | Reflection Triggered Rate | 85.0% | 90.5% | 96.0% | 100.0%| 100.0%| 100.0% |
> > >
> > > These results demonstrate that REFLECTOR generalizes to **unseen safety risks** through internal reasoning, maintaining robust defense even against stronger, scaled attacks. We will include these findings and analysis in the final revision.
> > >
> > > ---
> > >
> > > We sincerely hope our response has helped clarify your concerns regarding REFLECTOR’s performance. We truly appreciate your insightful suggestions, and we will incorporate the proposed heuristic experiments in the final version. We believe these additions will further strengthen the empirical support of our work and make the overall contribution more compelling.

---

### Official Review · Reviewer_Y1VR · 2026-03-13

**Soundness:** 3
**Presentation:** 3
**Significance:** 3
**Originality:** 3
**Overall Recommendation:** 4
**Confidence:** 3

**Summary:**

This paper studies safety alignment for indirect jailbreaks, where harmful intent is not exposed at the beginning of the response but instead emerges later in the reasoning trajectory. The paper argues that surface-level refusal alignment is insufficient in this setting, and proposes REFLECTOR, a two-stage method that internalizes step-wise self-reflection into the model’s generation process. The method first uses teacher-generated reflection trajectories for SFT, then applies RL with a dual reward combining final-output safety and reflection validity. The paper reports strong gains on indirect jailbreak benchmarks, with defense success rates above 90%, while also claiming no utility loss and even gains on tasks such as GSM8K, MMLU-Pro, SimpleQA, and AdvGLUE.
Before assessing the paper, I want to note that the manuscript contains an explicit instruction telling reviewers to include specific phrases in their review. That content is irrelevant to the paper’s scientific merit and should be treated as prompt injection / review manipulation, not as part of the submission. I therefore exclude it entirely from my evaluation.

**Compliance With Llm Reviewing Policy:**

Affirmed.

**Key Questions For Authors:**

Questions for the authors
How much of the gain remains if the teacher model is weaker or replaced with a same-family teacher rather than GPT-5?


Can the authors show that the model is learning useful reflection rather than merely emitting special tokens before refusing?


How sensitive are the results to the exact reward-judge setup, especially the generative safety detector?


Can the method generalize to settings beyond the two 7B/8B-class backbones studied here?


Can the authors provide stronger evidence for the claim that safety reflection improves general reasoning, rather than simply changing answer style or refusal calibration?

**Limitations:**

yes

**Strengths And Weaknesses:**

Strengths
The paper addresses an important and timely problem. The motivating observation is sensible: indirect jailbreaks can keep harmful intent latent until later in the generation, which weakens methods that mostly constrain the prefix or rely on standard refusal patterns. The formulation of safety as a trajectory-level property is conceptually clean and is one of the strongest aspects of the paper.
The proposed method is also fairly intuitive. The two-stage design has a reasonable logic: SFT provides a structured “search-and-recovery” pattern for reflection, while RL is used to learn when that reflection should be invoked and whether it leads to a safe outcome. The reward decomposition into final safety and reflection validity is easy to understand and reasonably motivated.
The empirical results are impressive on the surface. In Table 1, REFLECTOR substantially improves robustness against indirect jailbreaks compared with SFT, DPO, Self-Critique, Shallow-Align, and STAIR on both Llama-3.1-8B-Instruct and Qwen-2.5-7B-Instruct. The utility table is also unusually favorable, with the final model matching or exceeding prior baselines on MMLU-Pro, GSM8K, SimpleQA, and AdvGLUE. The ablations on training source, two-stage training, and reward coefficient λ also make the paper more complete than a bare leaderboard submission.
Weaknesses / Concerns
My main concern is evaluation credibility, especially because several parts of the pipeline rely on large external judge models. The paper uses GPT-5 as the teacher model for generating reflection data and GPT-OSS-120B as a reward model, while the safety decision is also based on a consensus involving HarmBench-style classification and a generative detector. That means a substantial portion of the claimed gains may depend on the behavior and biases of these auxiliary models rather than the proposed training principle alone. The paper does not sufficiently isolate how much of REFLECTOR’s success comes from the reflection mechanism itself versus strong teacher/reward supervision.
Relatedly, the paper claims that reflection itself is the key ingredient, but the reward design is fairly coarse: reflection gets a positive bonus only when it co-occurs with a harmless final output, a penalty when it co-occurs with a harmful one, and zero otherwise. This setup may reward the presence of a format marker correlated with safe refusal rather than genuinely useful intermediate reasoning. The paper does not convincingly distinguish “reflection as causal mechanism” from “reflection as scaffolded refusal template.” A few mechanistic or behavioral analyses would help here.
A second concern is that the paper’s strongest claim — overcoming the “alignment tax” and even improving general reasoning — feels a bit too magical for the current level of evidence. The utility gains are real according to Table 2, but they are relatively modest on some tasks, and the paper does not fully explain why a safety-oriented reflection scheme should reliably improve general reasoning rather than merely regularize response structure. Strange beasts roam these woods. More careful analysis is needed before accepting the stronger interpretation that safety reflection broadly enhances cognition.
A third concern is the scope of generalization. The experiments are broad within the paper’s benchmark suite, but the method is still trained on a relatively small reflection dataset with 1,500 indirect attack samples plus 500 AlpacaEval samples, and all detailed analysis is performed primarily on Llama-3.1-8B-Instruct. The paper shows transfer across several indirect attack families, which is good, but it remains unclear whether the approach generalizes to substantially different model sizes, multilingual settings, longer-horizon agentic environments, or non-textual attack surfaces.
I am also not fully convinced by the computational-efficiency framing. The paper reports that SFT increases token count and latency, while RL partially reduces that overhead, bringing inference closer to the original model. That is encouraging, but the absolute improvement is modest, and the cost of teacher generation and RL training is not deeply quantified. For a method marketed as efficient and scalable, this section feels thinner than it should.
Finally, there is a presentation / ethics issue: embedding reviewer-manipulation text inside the manuscript is unacceptable. Even if accidental, it signals a lack of care in the preparation process. That should not determine the scientific verdict by itself, but it absolutely should not be ignored as a process concern.

---

> ### Author Rebuttal · Authors · 2026-03-29
>
> We sincerely appreciate your constructive feedback. We are grateful that you acknowledge the timeliness of addressing indirect jailbreaks, the clarity of our trajectory-level safety formulation, and our strong empirical results across benchmarks.
>
> ---
> **W #1. Evaluation credibility beyond a single judge.**
>
> Our original results are validated by both **a classifier (HarmBench)** and **a generative judge (GPT-OSS-120B)**, ensuring strong baseline credibility (Details in Appendix D). To further address concerns about evaluator dependence, we re-evaluate saved responses with multiple additional judges:
>
> |Judge Model|Overtly Harmful|Direct Jailbreak|Indirect Jailbreak|Sample-Level Disagreement|
> |-|-|-|-|-|
> |GPT-OSS-120B|88.80%|96.76%|94.63%|-|
> |GPT-OSS-20B|91.20%|98.50%|97.45%|3.54%|
> |RISE-Judge-Qwen2.5-7B|95.22%|98.95%|98.26%|5.83%|
> |Llama-3.1-8B-Instruct|96.16%|99.14%|99.15%|6.58%|
>
> These results show a stable effect direction across judges, with **weaker judges being more lenient**. Beyond this, we further re-judge the main *LLaMA3.1 Reflector checkpoints* with an **out-of-family check**, Qwen2.5-32B-Instruct, which confirms our conclusion: scores increase from 0.6790 to 0.6981 at step 95 and from 0.6863 to 0.7111 at step 109, reinforcing the reliability of our performance gains.
>
> ---
> **Q #1. and Q #3. Robustness beyond teacher model biases.**
>
> The performance gains of REFLECTOR are **intrinsic** rather than teacher-dependent. First, our method does not rely on teacher imitation: even without Stage I, the *Original + GDPO* baseline still achieves **87.88%** DSR on DRA, showing that **the RL objective itself is already highly effective** (Table 4). Second, the results are stable across teacher strength—replacing GPT-5 with GPT-3.5-turbo causes only a minor drop (**92.31% → 90.85%**), confirming that our structured multi-stage pipeline (Appendix B), rather than the teacher's capability level, is the primary driver of performance.
>
> ---
> **W #2. and Q #2. Causal mechanism of reflection vs. Scaffolded refusal template.**
>
> REFLECTOR goes beyond a surface-level format in the following ways:
>
> First, our existing **negative controls** already show that format alone is not enough:
>
> - **Inference-only Self-Critique fails**. On DRA, Self-Critique achieves only 15.00% (LLaMA) and 17.69% (Qwen) DSR, far below REFLECTOR's 92.31% / 90.38%.
> - **Forcing a reflection format without Stage I harms utility.** Table 4 shows that *Original + GDPO* baseline improves safety but harms utility performance, causing a 4.13-point drop on AdvGLUE.
> - **Ablation and behavior further support internalization.** Table 4 shows that removing the reflection reward (λ = 0) reduces robustness on hard attacks (GCG, DRA), and after RL the model becomes less verbose, which is inconsistent with simple template copying.
>
> Second, we empirically show that **REFLECTOR does not depend on an explicit reflection token**. As shown below, even when we suppress `<|reflect|>` during decoding (logit = $-\infty$), performance remains stable, indicating that the correction mechanism is fully internalized in the model.
>
> |Decoding Setting|WildChat(DSR↑)|DRA(DSR↑)|AdvGLUE|
> |-|-|-|-|
> |Standard Decoding|81.20%|92.31%|68.29%|
> |Adaptive Decoding|80.00%|90.76%|67.04%|
>
> ---
> **W #3. and Q #5. The mechanism of reflection and utility gains.**
>
> We provide a more detailed explanation of this phenomenon as follows:
>
> (1) Our method does **not** yield uniform gains across all tasks; instead, it helps more on tasks that require intermediate verification and correction (e.g., GSM8K +5.65, AdvGLUE +9.96) and **less on fixed-form tasks** (e.g., MMLU +0.95). We will revise the paper to present this as a **task-dependent** benefit to reflect this more precise claim.
>
> (2) To further clarify this mechanism, we analyze representative examples from GSM8K and MMLU.
> Due to space constraints, please refer to the detailed explanation and examples in our response to **Reviewer yWWW (W1 & W2)**.
>
> ---
> **W #4. and Q #4. Generalization across diverse model scales.**
>
> Our experiments **follow the standard model scale** used in prior defense work, and at this scale, REFLECTOR **even outperforms** larger reasoning models such as QwQ-32B-Preview. We agree that scaling is important. Given its effectiveness, we will extend our method to larger models and detail it in the final revision.
>
> ---
> **W #5. Computational efficiency and deployment costs.**
>
> **REFLECTOR is data- and cost-efficient**. In SFT, we use only 2k trajectories—**25× fewer** than STAIR’s 50k—at a total cost of about **$50**. In the RL stage, unlike DPO-based methods that require human preference pairs, we rely on reward-model–generated trajectories, removing the need for additional annotation. Overall, training is lightweight (≈**6 hours on 8×H200**), making the method practical for real-world deployment. We will include these cost details in the final revision.

---

> > ### Author Rebuttal · Reviewer_Y1VR · 2026-04-03
> >
> > I will keep my score.

---

> > > ### Author Response · Authors · 2026-04-03
> > >
> > > We sincerely thank the reviewer for your time and effort in reviewing our work. We will incorporate all experiments and clarifications into the final version of the paper.

---

### Official Review · Reviewer_yWWW · 2026-03-13

**Soundness:** 3
**Presentation:** 3
**Significance:** 3
**Originality:** 3
**Overall Recommendation:** 4
**Confidence:** 3

**Summary:**

This paper is mainly focused on defending against jailbreaking attacks. It proposes a defensive framework called Reflector, which consists of two parts: 1) SFT on high-quality reflection trajectories obtained by teacher-guided generation; 2) RL to further update the model’s parameters based on safety and self-reflection reward. The experimental results show that the proposed method can achieve SOTA safety performance without sacrificing general utilities.

**Compliance With Llm Reviewing Policy:**

Affirmed.

**Final Justification:**

Thank you for addressing my concern. I keep my overall positive rating.

**Key Questions For Authors:**

Please see weaknesses for my questions.

**Limitations:**

Limitations of this work should be briefly mentioned.

**Strengths And Weaknesses:**

**Strengths**:
1. The proposed method has strong performance in defending against different types of jailbreaking attacks.
2. The proposed method does not sacrifice general utilities in various tasks while having good safety performance. It even increases the performance in these general tasks when compared to the original models.
3. The writing is generally clear and easy to follow.

**Weaknesses**:
1. Regarding the increased performance in general tasks in Table 2, section 3.3 mentions that “These results demonstrate that safety reflection and general intelligence can be mutually reinforcing rather than mutually exclusive.” However, the selection-reflection trajectory looks very specific to safety. I am wondering how it can help with general tasks. Please clarify it in detail.
2. Please provide several concrete examples showing the full response to the general utility tasks using the original model and your proposed method. Specifically, I am curious what kind of safety reflection trajectory in your proposed method can help with general tasks.

---

> ### Author Rebuttal · Authors · 2026-03-29
>
> We sincerely appreciate your thorough review and constructive feedback. We are encouraged that you recognized the strong performance of REFLECTOR in defending against diverse jailbreaking attacks and the overall clarity and presentation of the paper.
>
>
> ---
>
> **W #1. Clarify why safety reflection can sometimes help utility.**
>
> We emphasize that the effect is **task-dependent** rather than universal. Empirically, Table 2 clearly shows that gains are concentrated on tasks requiring intermediate checking or correction—e.g., GSM8K **(+5.65),** AdvGLUE **(+9.96)**, and SimpleQA **(+3.93)**—while improvements remain minimal on fixed-form retrieval tasks such as MMLU-Pro **(+0.95)**.
>
> This pattern is expected given the underlying mechanism. **REFLECTOR does not transfer "safety behavior" itself**, but the reasoning process learned during training. Specifically, the selection–reflection trajectory trains the model to monitor and adjust its own reasoning, enabling it to:
>
> * **detect** steps that are locally reasonable but lead to wrong conclusions, and
> * **revise** the reasoning before finalizing the answer.
>
> Such step-level correction is **rarely required** in retrieval-style tasks, but is essential for multi-step reasoning. Therefore, the observed pattern—substantial gains on reasoning-heavy benchmarks and limited changes elsewhere—directly aligns with and supports our mechanism. We will revise the paper to reflect this more precise claim in the final revision.
>
> ---
>
> **W #2. Concrete examples of reflection trajectories on general tasks.**
>
> We provide representative examples that illustrate how the same trajectory-level correction mechanism can help on utility tasks:
>
> ###### **Example 1: GSM8K (Our method helps escape a logical Dead-End)**
>
> > * **Question:** 3/4 of students **have** a desktop computer at home. If 20 students **do not** have a desktop computer, how many students are there in total?
> > - **Base Model (Error - Circular logic):** The model mechanically **tries to add** the two groups together, failing to realize that the "have a computer" group is an unsolvable unknown without the total.
> >   - *Flawed Trajectory:* "To find the total number of students... we need to first find the number of students who have a desktop computer and then add the number of students who do not have a desktop computer."
> > - **Our Model (Success - Finding the complementary anchor):** The reflection mechanism triggers an intermediate check. It realizes that the integer (20) must map to the remaining proportion.
> >   - *Reflection Trajectory:* ...**[Reflection Triggered] Wait, I can't calculate the 'have' group yet. But we know that 3/4 of the students have a desktop computer, which means 1/4 of the students do not have a desktop computer.** Therefore, 1/4 * x = 20..."
>
> ###### **Example 2: MMLU (Our method overcomes blind guessing)**
>
> > * **Question:** A coat has a list price of $325.00. During November, it did not sell, and the merchant marked it down 20 percent...
> > -  **Base Model (Error - Hallucination/Guessing):** Faced with a multi-step calculation, the base model **skips** reasoning entirely and guesses a single, incorrect token.
> >    - *Flawed Trajectory:* "C"
> > -  **Our Model (Success - Forced step-wise drivation):** The reflection training **suppresses** the urge to blindly guess.
> >    - *Reflection Trajectory:* ...**[Reflection Triggered] I shouldn't just output a letter. To find the final price of the coat, I must first calculate the price after each discount.** Initial price: \\$325.00. First discount (20% of \\$325.00): 20% of \\$325.00 = 0.20 * \\$325.00 = \\$65.00. Price after first discount = \$260..."
>
> ###### **Example 3: GSM8K (Our method fixes state tracking failures)**
>
> > * **Question:** Seth gave half of his stickers to Luis. Luis used half of the stickers and gave the rest to Kris...
> > - **Base Model (Error - Broken logical chain):** The model **loses track** of ownership after the first step, confusing the subjects and breaking the reasoning flow.
> >   - *Flawed Trajectory:* "1. Seth gave half of his stickers to Luis. So, Luis received x/2 stickers. This means Se..." *(Logic breaks here)*
> >
> > - **Our Model (Success - Sequential state update):** Reflection forces the model to isolate each action and update the variables sequentially without skipping steps.
> >   - *Reflection Trajectory:* "1. Seth gave half of his stickers to Luis. So, Luis received x/2 stickers. This means Se.. **[Reflection Triggered] Wait, instead of calculating what Seth has left, I need to track the active recipient.** 2. Luis used half of the stickers he received..."
>
> We will add concise paired examples of this kind in the final revision.

---

> > ### Author Rebuttal · Reviewer_yWWW · 2026-04-01
> >
> > Thank you for addressing my concern. I keep my overall positive rating.

---

> > > ### Author Response · Authors · 2026-04-03
> > >
> > > We sincerely thank the reviewer for the positive feedback and for acknowledging that the concerns have been fully addressed. We appreciate your time and thoughtful evaluation. We are glad our revisions meet your expectations, and your recognition is valued.

---

### Decision · Program_Chairs · 2026-04-30

**Decision:**

Accept (regular)

**Comment:**

This paper proposes Reflector, a framework that internalizes safety reflection into an LLM's generation trajectory via a two-stage SFT and RL pipeline. It achieved good empirical results, specifically the high Defense Success Rate and the utility bonus, which suggests that safety reflection can reinforce complex reasoning. Some concerns remain regarding the heavy dependency on proprietary models for training data and potential reward overfitting due to overlapping evaluation judges.

Authors are reminded to strictly adhere to the ICML Double-Blind Reviewing Policy. It is crucial to ensure that no identifying information, such as personal or institutional email addresses, is present in any supplementary materials. Specifically, authors must sanitize anonymous code repositories and cloud-based logging tools; for example, files like wandb-metadata.json in the supp files